# Evaluating and Rewarding LALMs for Expressive Role-Play TTS via Mean Continuation Log-Probability

Yong Ren [* 1 2 3]   Jingbei Li [* 3]   Haiyang Sun [3]   Yujie Chen [4]   Cheng Yi [3]   Yechang Huang [3]   Hao Gu [1 2]
Ye Bai [1]   Xuerui Yang [3]

## Abstract

Recent advances in Large Audio Language Models (LALMs) have extended Text-to-Speech (TTS) to interactive role-play scenarios, which demand high expressiveness and strict adherence to role-play instructions. However, existing models struggle to maintain stylistic consistency with character profiles and scene descriptions across multi-turn dialogues. A critical bottleneck is the lack of objective metrics for quantifying speaking style. To bridge this gap, we propose **Mean Continuation Log-Probability (MCLP)** as both an evaluation metric and a reward signal, validated on LALM-based Role-Play TTS (RP-TTS) tasks. MCLP leverages the in-context learning capability of pretrained LALMs to measure the likelihood of ground-truth speech tokens conditioned on a contextual history consisting of the transcript, generated speech, and repeated transcript, serving as a proxy for stylistic continuity. Furthermore, we employ MCLP as a reinforcement learning reward to enhance the style alignment between generated speech and role-play instructions. To support this task, we construct a large-scale RP-TTS dataset with rich scene and character annotations. Experiments demonstrate that MCLP is well aligned with human judgments of stylistic consistency and serves as an effective reward for improving RP-TTS, leading to consistent gains in both objective metrics and subjective evaluations. Our code is publicly available at https://github.com/y-ren16/MCLP.

## 1. Introduction

With the advancements in Large Language Models (LLMs), Text-to-Speech (TTS) has evolved from traditional objectives of intelligibility and acoustic fidelity to the generation of context-aware expressive speech with fine-grained control. Realizing this goal hinges on two pivotal capabilities: *timbre control* and *style control*. Timbre control is typically achieved through zero-shot voice cloning (Du et al., 2024; Zhou et al., 2025) or emerging voice design methodologies (Hu et al., 2026a;b). In contrast, style control remains a significant challenge.

Instruction-based TTS (Instruct-TTS) approaches (Vyas et al., 2023; Ji et al., 2025) leverage natural language prompts to control the target style of generated speech. While recent works have introduced open-ended textual descriptions to improve flexibility (Chen et al., 2026; Ren et al., 2026), these systems still struggle to handle complex multi-turn interactions, particularly in Role-Play TTS (RP-TTS) scenarios. Unlike recent Speech Role-Playing Agents (Jiang et al., 2025; Wu et al., 2025b), which primarily focus on semantic alignment with predefined role profiles, RP-TTS emphasizes stylistic consistency. This includes: (1) faithful alignment between the generated speech style and role-play instructions, such as scene and character descriptions, and (2) the preservation of stylistic coherence across multi-turn dialogues. A major bottleneck in improving stylistic consistency lies in the absence of objective evaluation metrics and effective reward signals. Supervised fine-tuning (SFT) alone is insufficient to meet the generalization requirements of RP-TTS. Existing Reinforcement Learning (RL) approaches for TTS style alignment often rely on emotion classification as a surrogate reward (Wang et al., 2025). However, emotion-based rewards fail to capture the rich, non-emotional stylistic attributes required for authentic role-play speech.

In this paper, we propose an interpretable stylistic consistency metric leveraging the In-Context Learning (ICL) capabilities of LALMs. We hypothesize that pretrained LALMs implicitly encode a continuous latent space of audio styles learned from large-scale speech corpora. Under this assumption, given a transcript and a candidate generated audio, a

---

[*]Equal contribution [1]Institute of Automation, Chinese Academy of Sciences [2]School of Artificial Intelligence, University of Chinese Academy of Sciences [3]StepFun [4]Beihang University. Correspondence to: Jingbei Li <lijb19@tsinghua.org.cn>, Cheng Yi <yicheng@stepfun.com>, Xuerui Yang <yangxuerui@stepfun.com>.

*Proceedings of the 43rd International Conference on Machine Learning*, Seoul, South Korea. PMLR 306, 2026. Copyright 2026 by the author(s).

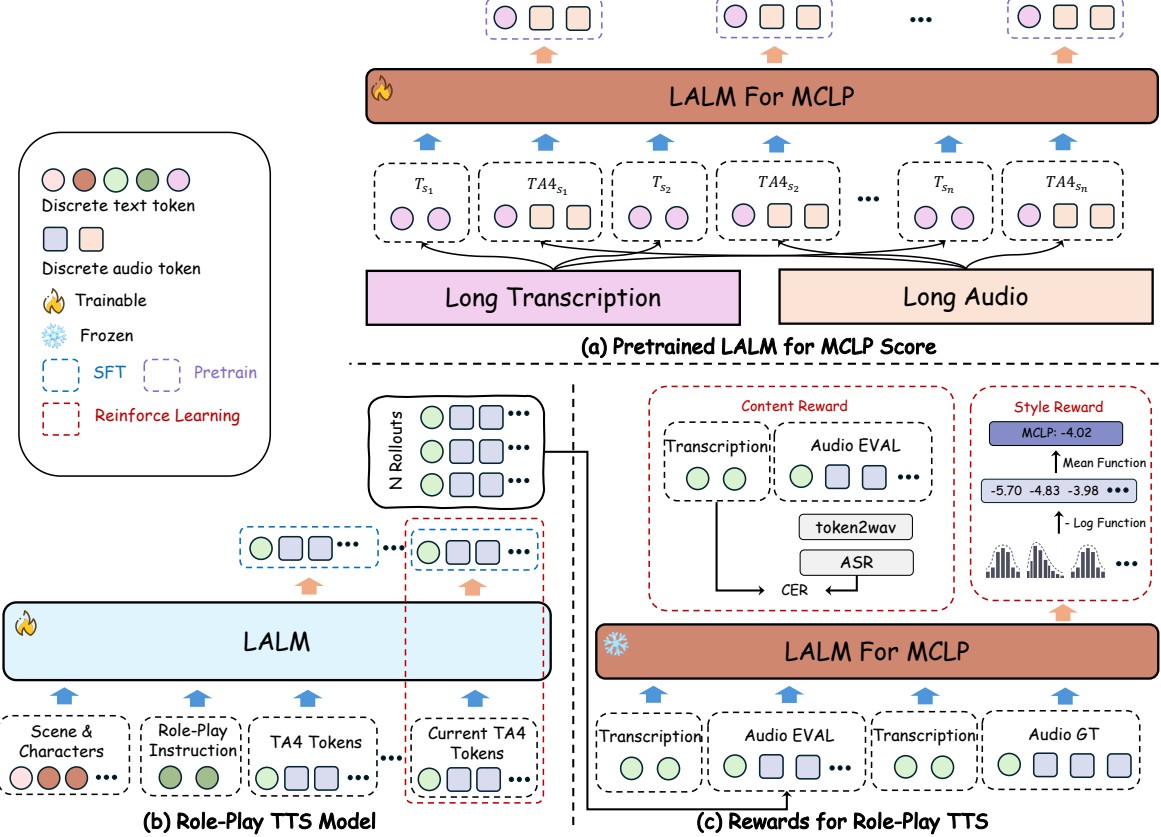

Figure 1. **The overall framework of our proposed method.** (a) The pretraining stage of the LALM for computing MCLP. (b) The LALM is fine-tuned to generate interleaved TA4 tokens conditioned on a structured prompt containing scene descriptions, character profiles, and dialogue history. (c) The hybrid reward function for GRPO that synergizes the style-centric MCLP signal with a content-fidelity penalty (CER) to prevent reward hacking.

LALM should assign higher likelihood to the ground-truth audio tokens of the same transcript when the underlying speaking style is consistent. Building on this insight, we propose Mean Continuation Log-Probability (MCLP), defined as the average log-probability of ground-truth (GT) audio tokens conditioned on the evaluation audio as contextual input. MCLP serves as a proxy for stylistic continuity in the latent style space: higher scores correspond to coherent stylistic continuation, whereas lower scores indicate stylistic divergence. Beyond evaluation, we further incorporate MCLP as a reward signal for optimizing RP-TTS systems. To prevent reward hacking, we construct a composite reward that combines MCLP with Character Error Rate (CER). We adopt Group Relative Policy Optimization (GRPO) (Shao et al., 2024) to jointly improve stylistic instruction adherence and content accuracy. Our contributions are summarized as follows:

- **Stylistic Consistency Metric MCLP**: We propose MCLP, an interpretable objective metric that quantifies stylistic consistency by leveraging the probabilistic priors of pretrained LALMs.

- **RL for RP-TTS**: We design a composite reward that integrates MCLP with CER and optimize RP-TTS models via GRPO, effectively improving stylistic alignment while preserving content fidelity.

- **Dataset & Experiments**: We construct a high-quality, multi-turn RP-TTS dataset with detailed annotations. Extensive experiments show that MCLP aligns well with human judgments of stylistic consistency and serves as an effective reward for improving RP-TTS performance across objective and subjective metrics.

## 2. Related Work

### 2.1. LLM-Based Controllable TTS

LLM-based TTS models (Chen et al., 2025; 2024; Du et al., 2024; Ye et al., 2025) have shown impressive zero-shot performance, particularly in timbre cloning via acoustic prompting. In contrast, controllable style generation—encompassing prosody, emotion, speaking manner, and expressive nuance—remains substantially underexplored. Earlier works (Guo et al., 2023; Leng et al., 2024;

Yang et al., 2024) defined style through discrete or coarse-grained attributes, whereas recent instruction-based methods (Yang et al., 2025; Zhou et al., 2025; Ren et al., 2026; Chen et al., 2026) adopt open-ended natural language descriptions to enhance expressiveness. However, these approaches largely model style as a static property tied to individual utterances, rather than as a temporally coherent and context-dependent performance. Unlike timbre, which is largely speaker-intrinsic, role-play style must evolve with conversational dynamics, scene transitions, and interaction history. Consequently, current instruction-based TTS systems struggle to sustain stylistic consistency over multi-turn dialogues, limiting their applicability to role-play TTS settings.

## 2.2. Speech Role-Playing Agents

To bridge the gap between static style control and dynamic character portrayal, recent research has pivoted towards Speech Role-Playing Agents. Fundamental to this progress is the construction of multimodal benchmarks derived from movies and games, such as OmniCharacter (Zhang et al., 2025b), SpeechRole (Jiang et al., 2025), VoxRole (Wu et al., 2025b), and AudioRole (Li et al., 2025). From a data standpoint, existing benchmarks either depend on character-centric synthetic data (Zhang et al., 2025b; Jiang et al., 2025) or remain limited in scale (Wu et al., 2025b), which constrains their ability to support comprehensive modeling of expressive and consistent role-play speech. From a modeling perspective, end-to-end speech-language architectures have emerged as the prevailing paradigm. While these systems jointly model role-play content and speech generation quality, they predominantly emphasize semantic role alignment over stylistic consistency in the acoustic domain. Consequently, despite improvements from supervised SFT, models often sacrifice acoustic expressiveness and stylistic appropriateness to preserve semantic coherence, particularly in multi-turn interactions (Shi et al., 2025). In contrast, our study focuses on Role-Play TTS under content-specified settings, where the response text is given, and the primary objective is to ensure that the generated speaking style remains consistent with the role-play instructions.

## 2.3. RL for TTS Alignment

To transcend the generalization limits of SFT, RL has been introduced to align TTS models with specific quality objectives (Du et al., 2025; Atamanenko et al., 2025; Gao et al., 2025a). However, RL performance is fundamentally constrained by the expressiveness of the reward function. While established metrics such as CER and speaker similarity effectively capture intelligibility and timbre, they fail to reflect stylistic coherence. Existing RL-based approaches often rely on emotion classification as a proxy for style (Liu et al., 2021; Gao et al., 2025b). Such discrete labels inadequately represent the continuous and context-dependent

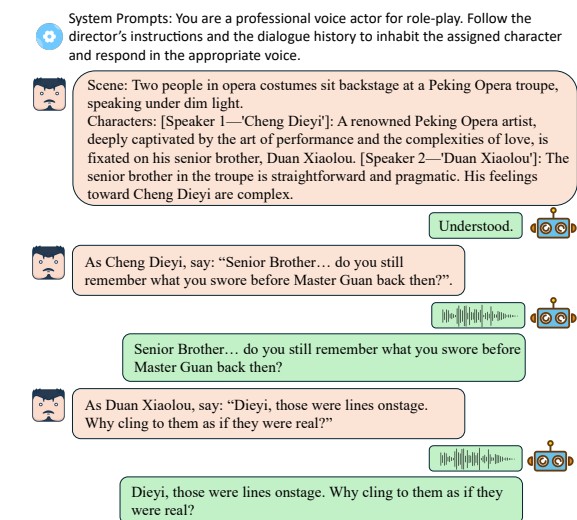

*Figure 2.* **An example for Role-Play TTS.**

nature of role-play style, limiting their applicability to expressive speech generation. In contrast, our proposed MCLP leverages the intrinsic likelihood modeling of pretrained LALMs to directly quantify stylistic consistency in latent space, providing a smooth, dense, and interpretable reward signal better suited for RL-based RP-TTS optimization.

## 3. Problem Preliminaries

We formulate RP-TTS as a context-aware conditional generation problem. Unlike standard TTS, which focuses on text-content fidelity, RP-TTS demands that the synthesized speech aligns with complex textual definitions regarding scene atmosphere and character persona, as illustrated in Figure 2. Formally, a dialogue session is grounded in a global context tuple $\mathcal{G} = (\mathcal{S}, \mathcal{P})$, where:

- $\mathcal{S}$ is the *Scene Description*, specifying the physical environment and ambient atmosphere.

- $\mathcal{P} = \{p_1, p_2, \ldots, p_K\}$ denotes the set of *Character Profiles* for $K$ distinct speakers, where $p_k$ encapsulates the personality and social role of the $k$-th character.

The generation process is dynamic depending on the turn index $t$. At the initial turn ($t = 1$), the model synthesizes speech conditioned solely on the semantic understanding of $\mathcal{G}$. For subsequent turns ($t > 1$), the model additionally incorporates the history of previous turns.

## 4. Methodology

### 4.1. Continuation Score

Speech style is deeply coupled with emotion, prosody, and paralinguistic factors, making it difficult to define and mea-

sure objectively. With the scaling of model parameters and dataset sizes, LALMs pretrained on massive speech corpora have demonstrated strong ICL capabilities. Capitalizing on this phenomenon, we propose MCLP, an objective metric that leverages a pretrained LALM to quantify the stylistic consistency between a candidate speech utterance and a ground-truth speech reference.

**LALM Model Choice.** MCLP is computed using a generative LALM initialized from Step-Audio-2 (Wu et al., 2025a) and further trained as a continuation model, as illustrated in Figure 1(a). We choose this model because Step-Audio-2 uses a semantic speech tokenizer, which mainly preserves semantic and stylistic information rather than acoustic details. Under the fixed-transcript setting of MCLP, this design helps bias the metric toward stylistic consistency rather than acoustic similarity.

**Continuation Training.** To strengthen the continuation modeling capability used by MCLP, we further train the Step-Audio-2-based LALM on 3 million hours of transcribed speech. For each continuous speech session $\mathcal{S} = \{s_1, s_2, \ldots, s_n\}$, we construct a session-level sequence $\{\mathbf{w}_1, \mathbf{y}_1, \mathbf{w}_2, \mathbf{y}_2, \ldots, \mathbf{w}_n, \mathbf{y}_n\}$, where $\mathbf{w}_i$ is the transcript and $\mathbf{y}_i$ is the corresponding TA4 token sequence of sentence $s_i$. The model is trained autoregressively, with the loss applied only to the interleaved audio tokens in each $\mathbf{y}_i$. This enables the model to predict speech conditioned on both textual content and preceding speech context.

**Definition of MCLP.** Let $P_\theta$ denote the autoregressive distribution over TA4 tokens parameterized by the continuation model. During evaluation, as illustrated in Figure 1(c), we assess a candidate audio sequence $\mathbf{z}^{eval}$ against a ground-truth reference $\mathbf{z}^{gt}$ for the same transcript $\mathbf{w}$. We construct a dual-turn context history $\mathcal{H} = [\mathbf{w}, \mathbf{z}^{eval}, \mathbf{w}]$, where the transcript is repeated before the target continuation. This design fixes the linguistic content and teacher-forces text tokens, so the variation in continuation likelihood mainly reflects whether $\mathbf{z}^{eval}$ provides a compatible speaking style for predicting $\mathbf{z}^{gt}$. Although this order may appear counterintuitive, the objective is to measure stylistic consistency between $\mathbf{z}^{eval}$ and $\mathbf{z}^{gt}$, which is conceptually symmetric. We predict $\mathbf{z}^{gt}$ from $\mathbf{z}^{eval}$ because the ground-truth length is fixed, allowing natural normalization and fair comparison among multiple candidates evaluated against the same reference. We compute MCLP as the mean log-likelihood of the ground-truth audio tokens conditioned on this history:

$$\text{MCLP}(\mathbf{z}^{eval}, \mathbf{z}^{gt}) = \frac{1}{|\mathbf{z}_A^{gt}|} \sum_{k \in \mathbf{z}_A^{gt}} \log P_\theta(z_k^{gt} \mid \mathcal{H}, z_{<k}^{gt}), \quad (1)$$

where $\mathbf{z}_A^{gt}$ denotes the subset of audio tokens in the ground-truth TA4 sequence.

### 4.2. Supervised Fine-Tuning

We perform a SFT procedure (Wei et al., 2022) to instruct the LALM, Step-Audio-2-mini-Base (Wu et al., 2025a), to adhere to scene and character constraints in RP-TTS tasks. We utilize the dataset $\mathcal{D}_{rp}$ constructed in Section 5. To capture the multi-turn dynamics, we decompose each dialogue session into turn-level training samples. For a target utterance at turn $j$, we define the conditional context as a concatenation of the global setup and the dialogue history. Formally, let $\mathcal{S}$ be the scene description, $\mathcal{P}$ be the character profiles, and $\mathcal{I}_j$ be the specific instruction for the $j$-th turn (containing the transcript $\mathbf{w}_j$). Let $\mathcal{H}_{<j} = \{(\mathcal{I}_k, \mathbf{y}_k)\}_{k=1}^{j-1}$ represent the history of previous turns. The target $\mathbf{y}_j$ is the interleaved TA4 sequence. As illustrated in Figure 1(b), the model is optimized to minimize the negative log-likelihood of the target sequence given the full context:

$$\theta^* = \arg \min_\theta \sum_{(\mathcal{S}, \mathcal{P}, \mathcal{H}, \mathcal{I}, \mathbf{y}) \in \mathcal{D}_{rp}} - \log P_\theta(\mathbf{y} \mid \mathcal{S}, \mathcal{P}, \mathcal{H}, \mathcal{I}), \quad (2)$$

where $\theta$ represents the trainable parameters of the LALM. Through this objective, the model learns to synthesize speech that is aligned with the dialogue history and the designated scene and character.

### 4.3. Reinforcement Learning

We further align the RP-TTS model using GRPO on a curated high-quality subset, described in Section 5.3. As illustrated in Figure 1(b, c), the RL alignment is exclusively applied to the synthesis of the final turn. For each query $\mathbf{q}$ (comprising the scene, profiles, and dialogue history), we sample a group of $G$ rollouts $\{\mathbf{o}_1, \ldots, \mathbf{o}_G\}$ from the old policy $\pi_{\theta_{\text{old}}}$. The objective function is formulated as:

$$\mathcal{J}_{\text{GRPO}}(\theta) = \mathbb{E}_{\substack{\mathbf{q} \sim P(\mathbf{q}) \\ \{\mathbf{o}_i\} \sim \pi_{\theta_{\text{old}}}}} \left[ \frac{1}{G} \sum_{i=1}^{G} \frac{1}{|\mathbf{o}_i|} \sum_{t=1}^{|\mathbf{o}_i|} \left( \min \left[ \rho_{i,t} \hat{A}_i, \right. \right. \right.$$
$$\left. \left. \left. \text{clip}(\rho_{i,t}, 1 - \epsilon, 1 + \epsilon) \hat{A}_i \right] - \beta \mathbb{D}_{\text{KL}} \right) \right], \quad (3)$$

where $\rho_{i,t} = \frac{\pi_\theta(o_{i,t} | \mathbf{q}, \mathbf{o}_{i,<t})}{\pi_{\theta_{\text{old}}}(o_{i,t} | \mathbf{q}, \mathbf{o}_{i,<t})}$ is the probability ratio, and $\mathbb{D}_{\text{KL}}$ represents the token-level KL divergence between the policy and the reference model SFT. The advantage $\hat{A}_i$ is computed by normalizing the rewards within the group to reduce variance:

$$\hat{A}_i = \frac{R_i - \text{mean}(\{R_1, \ldots, R_G\})}{\text{std}(\{R_1, \ldots, R_G\})}. \quad (4)$$

**Hybrid Reward Formulation.** A fundamental challenge in RP-TTS lies in balancing the trade-off between expressiveness and intelligibility. Naive optimization of a single

*Table 1.* **Dataset statistics across the processing pipeline.** The table tracks the volume (Hours, Sentences) and richness (Avg. Scenes/Speakers) of the data at each stage of filtering and annotation. The final dataset focuses on high-quality drama scenes with rich multi-turn interactions.

| Stage | #Audio | Sum sent. dur. (h) | #Sentences | #Scenes | Avg scenes/audio | Avg sent./scene | Avg spk./scene |
|---|---|---|---|---|---|---|---|
| WenetSpeech | 83,503 | 12,541.61 | 17,894,974 | - | - | - | - |
| YouTube Drama | 17,253 | 5,343.24 | 8,306,291 | - | - | - | - |
| YouTube Drama (video avail.) | 8,556 | 2,709.98 | 4,178,049 | - | - | - | - |
| Scene & Character annotated | 8,278 | 2,288.79 | 3,581,488 | 598,701 | 72.32 | 5.98 | 1.74 |
| Filtered | 8,237 | 1,435.29 | 2,276,602 | 311,938 | 37.87 | 7.30 | 2.33 |

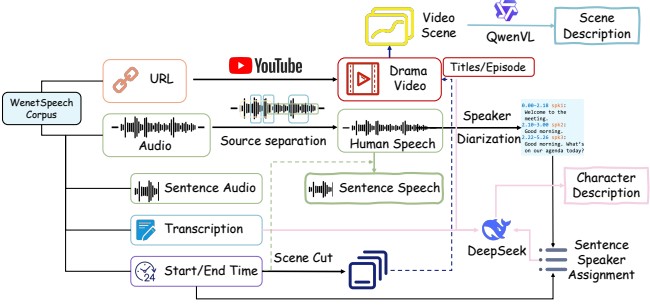

*Figure 3.* **Data Curation and Annotation Pipeline.** We construct the RP-TTS dataset from the WenetSpeech corpus via a multi-stage process.

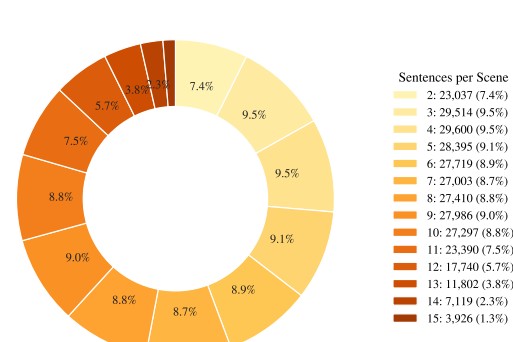

*Figure 4.* **Distribution of dialogue turns.** The number of utterances per scene in our RP-TTS dataset.

objective is prone to reward hacking (Skalse et al., 2022): over-optimizing for style frequently compromises semantic adherence, whereas focusing exclusively on content fidelity typically yields prosodically flat and unexpressive speech. To reconcile these conflicting objectives, we design a composite reward function $R(\mathbf{z})$ that synergizes the style-centric MCLP with a strict content fidelity constraint.

- **Style Reward** ($R_{style}$): We directly utilize the MCLP score with a bias term $C$ to quantify stylistic consistency:

$$R_{style} = \text{MCLP}(\mathbf{z}^{roll}, \mathbf{z}^{gt}) + C, \qquad (5)$$

where $C$ shifts the MCLP into a positive range.

- **Content Constraint** ($R_{content}$): We employ a cascaded Token-to-Wav module (Wu et al., 2025a) and Automatic Speech Recognition (ASR) model (Huang et al., 2025) to transcribe the generated audio tokens $\mathbf{z}^{roll}$ back into text $\hat{\mathbf{w}}$ and compute the CER against the ground truth $\mathbf{w}$. The content penalty is defined as:

$$R_{content} = \lambda \cdot \text{CER}(\hat{\mathbf{w}}, \mathbf{w}), \qquad (6)$$

where $\lambda$ is a penalty coefficient.

- **Gated Aggregation:** The final reward incorporates a gating mechanism to prevent the model from learning "expressive gibberish." If the CER exceeds a tolerance threshold $\tau$, the reward is zeroed out:

$$R(\mathbf{z}) = \begin{cases} 0 & \text{if } \text{CER}(\hat{\mathbf{w}}, \mathbf{w}) > \tau, \\ R_{style} - R_{content} & \text{otherwise.} \end{cases} \qquad (7)$$

This formulation effectively creates a curriculum: the model must first satisfy the strict intelligibility constraint before it can optimize for the stylistic gains offered by $R_{style}$.

## 5. Dataset Construction

To evaluate MCLP and support RP-TTS training, we construct WenetSpeech-RP-TTS[1], a large-scale role-play TTS dataset derived from real-world Chinese drama videos.

### 5.1. Data Acquisition and Preprocessing

As illustrated in Figure 3, we leverage the WenetSpeech corpus (Zhang et al., 2022) as our primary data source. To align with our role-play objective, we filter for subsets sourced from "YouTube" with the "drama" tag, resulting in 17,253 audio files. We successfully retrieved 8,556 valid videos based on available URLs. Subsequently, we employ DeepSeek-R1 (Guo et al., 2025) to infer series titles and episode numbers from video titles. Then we apply Demucs (Rouard et al., 2023) for source separation to isolate clean vocal tracks. For speaker identification, we perform speaker diarization (SD) on the whole audio using pyannote (Bredin, 2023). To achieve sentence-level speaker assignment, we

---

[1] https://huggingface.co/datasets/y-ren16/WenetSpeech-RP-TTS

align the timestamp of each transcript segment with the RTTM speaker segments.

## 5.2. Scene and Character Annotation

To instantiate the global context tuple $\mathcal{G} = (\mathcal{S}, \mathcal{P})$, we employ a hierarchical annotation strategy:

- **Character Annotation** ($\mathcal{P}$): Following SD, we aggregate all utterances associated with each unique speaker ID to reconstruct their dialogue history. Then we construct a global context containing the drama's title, episode and the dialogue of all characters. We feed this holistic script into DeepSeek-R1, prompting it to infer detailed character descriptions by analyzing the linguistic patterns and inter-character dynamics.

- **Scene Segmentation and Annotation** ($\mathcal{S}$): We partition the continuous audio stream into discrete dialogue scenes: a new scene is triggered by a silence gap exceeding 5 seconds, with a maximum duration cap of 30 seconds. For each scene, we employ the Qwen-VL-7B (Bai et al., 2025) to generate the physical environment and atmosphere descriptions of the video.

## 5.3. Dataset Details

**Dataset Statistics.** As illustrated in Table 1, the final processed corpus, encompassing both training and evaluation splits, comprises approximately 311k scenes, totaling 1,435 hours of high-fidelity speech. On average, each scene consists of 7.3 utterances involving 2.33 distinct speakers, providing a dense and diverse multi-turn environment for modeling stylistic consistency and interaction dynamics.

**Test Set Construction.** To strictly prevent data leakage and ensure fair evaluation, we implemented a video-level split. From the pool of 8,237 processed videos, we held out 200 videos as test candidates, ensuring no overlap in source material between training and evaluation. From these candidates, we constructed a balanced test set of 900 scenes via stratified sampling: we selected exactly 100 samples for each dialogue length ranging from 2 to 10 turns.

**RL Dataset Filtering.** For the RL stage, we curated a higher-quality subset from the SFT data to focus specifically on expressive synthesis. We applied the following criteria: 1) **Turn Constraint**: Dialogue length is restricted to 2–6 turns to focus on immediate context modeling; 2) **Content Length**: The text transcription of the final turn must exceed 10 Chinese characters to ensure sufficient prosodic complexity; 3) **Style Filtering**: We utilized an internal style classification model to filter out "Neutral" speech, retaining only samples classified as "Not Neutral" (i.e., highly emotional or stylized). This filtering process yielded 16,186

scenes, serving as the high-quality data for GRPO. Detailed prompts and additional dataset analyses are provided in Appendices A and C.

## 6. Experiments

### 6.1. Experimental Setup

**Model Configuration.** We employ Step-Audio-2-mini-Base (Wu et al., 2025a) as our base model. It is a 7B-parameter decoder-only LALM that processes interleaved text and audio tokens within a unified sequence.

**Baselines.** We first evaluate our method against three competitive LALMs that support multi-turn audio history as input and exhibit strong capabilities in both audio understanding and high-fidelity generation. These models are closely aligned with the target setting of RP-TTS, where the generated speech is expected to follow the scene and character descriptions while remaining stylistically coherent with previous dialogue turns.

- **GPT-Audio**: An advanced closed-source model developed by OpenAI, widely regarded as a strong baseline for audio comprehension and generation.

- **MiMo-Audio-7B-Instruct** (Zhang et al., 2025a): A recently proposed open-source LALM designed for end-to-end speech interaction. It is particularly noted for its powerful context-aware speech continuation and generation capabilities.

- **Step-Audio-2-mini**: The instruction-tuned variant of our backbone model, optimized for general speech tasks.

In addition, we include four recent Instruct-TTS systems: CosyVoice3 (Du et al., 2025), Higgs Audio V2 (Boson AI, 2025), OV-InstructTTS (Ren et al., 2026), and Qwen3TTS (Hu et al., 2026a). As they do not support multi-turn audio history, we evaluate them only in the w/o audio-history setting by rewriting the scene description, character profile, and target transcript into each model's required prompt format.

**Training Details.** Our training pipeline consists of two stages:

- **SFT Stage**: We fine-tune the base model for 1 epoch with a global batch size of 64. The learning rate is set to $1 \times 10^{-5}$ with cosine decay, 100 warmup iterations, and a minimum learning rate of $1 \times 10^{-6}$. We use AdamW with $\beta_1 = 0.9$, $\beta_2 = 0.95$, weight decay 0.1, and gradient clipping at 1.0. The maximum sequence length is 16,384.

- **RL Stage**: We initialize the actor and reference models from the SFT checkpoint and optimize the actor with

GRPO for 1,000 iterations. We use a learning rate of $1 \times 10^{-6}$ with 50 warmup iterations, a global batch size of 128, and sample $G = 8$ responses per prompt with temperature 1.0 and a maximum decoding length of 1,024. The KL coefficient is set to $\beta = 0.001$. For the reward parameters, we set $C = 15.0$, $\lambda = 10.0$, and $\tau = 0.2$. All RL experiments use 32 NVIDIA H800 GPUs.

**Evaluation Metrics.** To comprehensively assess the performance of Role-Play TTS, we employ a combination of objective and subjective metrics.

- **Objective Metrics**: We use CER to measure content fidelity, where generated speech is transcribed by a Step-Audio (Huang et al., 2025) model specifically fine-tuned on ASR tasks and compared with the target transcript. Since objective metrics for stylistic consistency remain underexplored, we also report two similarity-based proxy metrics for a more comprehensive evaluation: CAM++ (Wang et al., 2023; Zheng et al., 2023)[2] for speaker-embedding similarity, whose representations may leak style-related cues, and Emo2Vec (Ma et al., 2024)[3] for emotion-representation similarity, as emotion is an important component of speaking style.

- **Subjective Metrics**: We conduct subjective evaluation with 50 trained native Chinese professional annotators. The evaluation set contains 30 test cases: 10 without audio history and 20 with audio history, stratified into 8, 6, 4, and 2 cases for 2-, 3-, 4-, and 5-turn dialogues. For each case, annotators evaluate all applicable methods; Instruct-TTS baselines are evaluated only without audio history. We insert one explicit "gold standard" case at a random position for quality control, where compared samples exhibit an obvious stylistic difference. Annotators inconsistent on this case are filtered out, yielding 48 valid responses. We use a 0.5–5.0 MOS scale with 0.5 increments. Unlike conventional naturalness MOS, our MOS measures stylistic consistency and role-play fidelity. Annotators are instructed to review scene descriptions and character profiles, focus on target-style consistency and role portrayal, and disregard irrelevant timbre or audio-quality differences.

### 6.2. Validation of MCLP with Human Judgments

To assess whether the proposed MCLP aligns with human judgments of stylistic consistency, we conducted a separate correlation analysis with 32 expert listeners. We

[2] https://modelscope.cn/models/iic/speech_campplus_sv_zh-cn_3dspeaker_16k/summary
[3] https://huggingface.co/emotion2vec/emotion2vec_plus_large

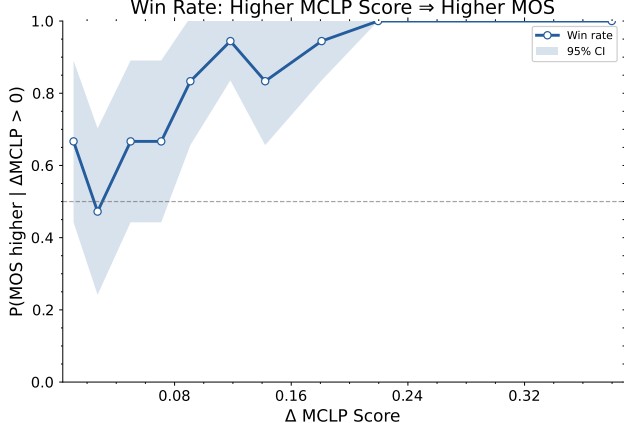

*Figure 5.* **Win rate vs. ΔMCLP Score.** The win rate (probability that a higher MCLP score predicts a higher human MOS) across different ΔMCLP bins.

performed pairwise comparisons by computing the differences $\Delta\text{MCLP} = \text{MCLP}_i - \text{MCLP}_j$ and $\Delta\text{MOS} = \text{MOS}_i - \text{MOS}_j$ for all possible pairs $(i, j)$. A pair was considered a win if the utterance with a higher MCLP score also received a higher human MOS score. These pairs were grouped into ten bins according to their ΔMCLP values. Figure 5 reports the average win rate for each bin, together with the corresponding 95% confidence intervals. The results reveal a clear and interpretable trend. When ΔMCLP is small, the win rate is close to 0.5, indicating performance comparable to random guessing. As ΔMCLP increases, the probability that the utterance with a higher MCLP score is also preferred by human listeners rises steadily. In particular, when $\Delta\text{MCLP} > 0.1$, the win rate exceeds 0.8, demonstrating a strong alignment between MCLP and human judgments of stylistic consistency.

### 6.3. Experimental Results

Table 2 presents the comprehensive performance comparison between our proposed method and competitive baselines under two settings: **(1) W/ Audio History**, where the model is provided with scene descriptions, character profiles, target transcripts, and historical audios; and **(2) W/o Audio History**, where the model is provided only with scene descriptions, character profiles, and target transcripts. LALM baselines are evaluated under both settings, whereas Instruct-TTS baselines are evaluated only under the w/o audio-history setting, since they do not support multi-turn audio history as input. Overall, our method consistently outperforms existing baselines in both content fidelity and stylistic instruction following across the two settings.

**Content Instruction Following.** We first evaluate the capability of different models to strictly adhere to the target transcript within the RP-TTS framework. As shown in

*Table 2.* Main performance comparison with baselines. **MOS** is reported with 95% confidence intervals. '◇' indicates that samples were resynthesized via the Step-Audio-2 tokenizer using ground-truth audio prompts to disentangle style from timbre and audio quality during subjective evaluation. **Red** indicates the best result, and **Blue** indicates the second best.

| Model | W/ Audio History | | | | W/o Audio History | | | | Subjective |
|---|---|---|---|---|---|---|---|---|---|
| | CER (%)↓ | CAM++↑ | Emo2Vec↑ | MCLP↑ | CER (%)↓ | CAM++↑ | Emo2Vec↑ | MCLP↑ | MOS↑ |
| *Ground Truth* ◇ | - | - | - | - | - | - | - | - | 4.461 (± 0.041) |
| *Multi-turn LALM Systems (with audio history support)* | | | | | | | | | |
| GPT-Audio ◇ | 11.974 | 0.636 | 0.875 | -4.849 | 44.679 | 0.635 | 0.884 | -4.836 | 1.915 (± 0.048) |
| MiMo-Audio-7B ◇ | 10.605 | **0.699** | **0.902** | **-4.753** | 11.609 | **0.698** | 0.903 | **-4.745** | 2.484 (± 0.058) |
| Step-Audio-2-mini | **3.276** | 0.629 | 0.864 | -4.829 | 12.007 | 0.632 | 0.864 | -4.823 | 1.856 (± 0.045) |
| *Instruct-TTS Systems* | | | | | | | | | |
| CosyVoice3 ◇ | - | - | - | - | 4.638 | 0.651 | **0.905** | -4.782 | 2.350 (± 0.078) |
| Higgs Audio V2 ◇ | - | - | - | - | **3.250** | 0.614 | 0.856 | -4.827 | 1.750 (± 0.064) |
| OV-InstructTTS | - | - | - | - | 7.188 | 0.669 | 0.900 | -4.768 | **2.864** (± 0.087) |
| Qwen3TTS ◇ | - | - | - | - | 5.585 | 0.630 | 0.879 | -4.799 | 2.036 (± 0.073) |
| **Our Proposed** | **1.130** | **0.724** | **0.917** | **-4.636** | **1.625** | **0.704** | **0.910** | **-4.687** | **3.576** (± 0.045) |

Table 2, among the LALM baselines, *Step-Audio-2-mini* achieves better content fidelity than *GPT-Audio* and *MiMo-Audio-7B-Instruct* in the w/ audio-history setting. However, a critical challenge in RP-TTS is that models may fail to follow transcription instructions precisely due to interference from rich contextual information. This failure mode is particularly pronounced in the "cold-start" setting without audio history; for instance, *GPT-Audio* suffers a severe degradation in intelligibility, and *Step-Audio-2-mini* also exhibits a clear performance drop. The single-turn controllable TTS baselines achieve reasonable CER, but they still lag behind our method. By incorporating a CER-based content constraint during RL, our method achieves the lowest CER in both settings, demonstrating strong robustness against contextual interference while preserving content fidelity.

**Stylistic Instruction Following.** Stylistic consistency and role-play fidelity are the central objectives of RP-TTS, and thus subjective score serves as the most important evaluation criterion. As shown in Table 2, our method achieves the highest MOS score of 3.576, substantially outperforming both the best multi-turn LALM baseline, *MiMo-Audio-7B* (2.484), and the best Instruct-TTS baseline, *OV-InstructTTS* (2.864). Since our MOS protocol asks annotators to focus on target-style consistency and role portrayal rather than naturalness, timbre, or audio quality, this result directly shows better alignment with scene descriptions, character profiles, and dialogue context.

We further analyze stylistic consistency using similarity-based proxy metrics. Since effective objective metrics for stylistic consistency were largely unavailable before MCLP, CAM++ and Emo2Vec provide complementary but partial perspectives. CAM++ primarily measures speaker-embedding similarity, whose representations may also encode style-related cues such as speaking prosody, while Emo2Vec measures emotion-representation similar-

ity, capturing an important aspect of speaking style. Our method achieves the best CAM++ and Emo2Vec scores in both settings, reaching 0.724/0.917 with audio history and 0.704/0.910 without audio history.

MCLP provides a more task-aligned objective measure of such stylistic coherence. Our method achieves the highest MCLP score among all comparable systems in both settings, improving over the strongest LALM baseline *MiMo-Audio-7B* from -4.753 to -4.636 with audio history and from -4.745 to -4.687 without audio history. This trend is consistent with the subjective MOS ranking, where our method also receives the best human preference.

Comparing different baseline categories further highlights the difficulty of RP-TTS. Multi-turn LALMs can accept audio history and therefore are closer to the target RP-TTS setting, but general-purpose models still struggle to translate dialogue history and role-play instructions into expressive and consistent speech. Among them, *MiMo-Audio-7B* shows the strongest stylistic adherence, outperforming *GPT-Audio* and *Step-Audio-2-mini* on MCLP, CAM++, Emo2Vec, and MOS. In contrast, single-turn Instruct-TTS systems can follow natural-language style prompts and sometimes achieve competitive proxy similarity scores, but they cannot directly condition on multi-turn audio history. Their evaluation is therefore limited to the w/o audio-history setting, where the scene description, character profile, and target transcript are rewritten into each model's prompt format. Despite this favorable single-turn approximation, our method still outperforms all Instruct-TTS baselines by a large margin in MOS and MCLP, demonstrating the benefit of training and aligning a model specifically for RP-TTS.

Finally, audio history provides an additional advantage for our model. Our MCLP improves from -4.687 in the w/o audio-history setting to -4.636 in the w/ audio-history setting. This confirms that our model can effectively exploit

*Table 3.* Ablation studies on training stages and reward components. **Red** indicates the best result, and **Blue** indicates the second best.

| Model | W/ Audio History | | | | W/o Audio History | | | | Subjective |
|---|---|---|---|---|---|---|---|---|---|
| | CER (%)↓ | CAM++↑ | Emo2Vec↑ | MCLP↑ | CER (%)↓ | CAM++↑ | Emo2Vec↑ | MCLP↑ | MOS↑ |
| Step-Audio-2-mini | 3.276 | 0.629 | 0.864 | -4.829 | 12.007 | 0.632 | 0.864 | -4.823 | 1.856 (± 0.045) |
| Step-Audio-2-mini (SFT) | 3.334 | 0.664 | 0.901 | -4.725 | 6.306 | 0.667 | 0.899 | -4.731 | 3.178 (± 0.052) |
| **Our Proposed** | 1.130 | 0.724 | 0.917 | -4.636 | 1.625 | 0.704 | 0.910 | -4.687 | 3.576 (± 0.045) |
| – w/o CER Reward | 61.144 | 0.677 | 0.914 | -4.590 | 59.665 | 0.621 | 0.910 | -4.637 | 1.145 (± 0.057) |
| – w/o MCLP Reward | 0.783 | 0.687 | 0.898 | -4.752 | 0.836 | 0.670 | 0.886 | -4.773 | 2.331 (± 0.046) |

historical acoustic cues to maintain dialogue-level stylistic continuity. In comparison, baseline models fail to fully benefit from audio history. These observations demonstrate that the proposed SFT and MCLP-based RL alignment not only improve isolated utterance-level expressiveness, but also enhance the ability to generate context-consistent role-play speech across multi-turn interactions. We provide additional fine-grained analyses by scene category, dialogue length, and speaker count in Appendix C.

## 6.4. Ablation Studies

To deconstruct the contribution of SFT, RL, and each component in our reward formulation, we conduct ablation studies as shown in Table 3.

- **SFT.** After performing SFT on *Step-Audio-2-mini-Base* using our curated dataset WenetSpeech-RP-TTS, the model obtains clear improvements in stylistic instruction following over the original *Step-Audio-2-mini*, as reflected by higher CAM++, Emo2Vec, MCLP, and MOS scores.

- **RL.** Applying RL with the proposed hybrid reward further improves both content fidelity and stylistic consistency. Compared with the SFT model, CER decreases by 2.204% and 4.681% in the w/ and w/o audio-history settings, respectively, while MCLP improves from -4.725 to -4.636 and from -4.731 to -4.687. The MOS also increases from 3.178 to 3.576, confirming that the RL stage improves perceptual stylistic consistency rather than merely optimizing the automatic metric.

- **Rewards.** The ablation results also demonstrate that optimizing a single reward objective makes the model prone to reward hacking. When optimizing solely for MCLP (*w/o CER Reward*), we observe severe hacking manifested as fixed repetitive acoustic patterns at the end of utterances. Consequently, the model achieves the highest MCLP score (-4.590) but suffers a catastrophic breakdown in intelligibility, with CER increasing to over 50%. This confirms that without the content constraint, the model can exploit the style reward by generating linguistically meaningless but acoustically repetitive outputs. Conversely, when optimizing solely for CER (*w/o MCLP Reward*), the model achieves the lowest CER (0.783%) but substantially degrades in stylistic consistency. Its MCLP score

drops to -4.752 in the w/ audio-history setting, and the MOS decreases to 2.331, far below the full model's 3.576. Qualitatively, the generated speech becomes flatter and less expressive, indicating that content fidelity alone is insufficient for RP-TTS. Our proposed method, by synergizing CER and MCLP into a hybrid reward, strikes a better balance between intelligibility and expressiveness, maintaining low CER while achieving substantially stronger objective and subjective stylistic consistency.

## 7. Conclusion

In this work, we addressed the critical challenge of maintaining stylistic consistency in Role-Play TTS. We identified the lack of explainable style metrics as the primary bottleneck and proposed MCLP as a novel solution. By leveraging the in-context learning priors of LALMs, MCLP quantifies style as the "reverse continuation likelihood" of the ground truth, effectively transforming abstract style into a tractable metric. We integrated this metric into a GRPO-based RL framework, designing a gated hybrid reward that harmonizes stylistic expressiveness with linguistic accuracy. Experiments on WenetSpeech-RP-TTS show that MCLP aligns well with human judgments and functions as an effective reward signal, enabling our method to outperform strong LALM and Instruct-TTS baselines across objective and subjective evaluations.

## 8. Limitations

While our work establishes MCLP as an effective stylistic consistency metric and reward for RP-TTS, it has several limitations. First, the current WenetSpeech-RP-TTS dataset and main evaluation are limited to Mandarin. Although preliminary results in Appendix B suggest that MCLP can generalize to English, comprehensive multilingual validation remains future work. Second, while our dataset covers diverse drama genres, further evaluation is needed on broader domains such as audiobooks, games and virtual assistants. Third, the scene and character descriptions are automatically generated by LLMs, which may introduce annotation noise or bias despite filtering and manual inspection.

## Impact Statement

This paper advances expressive speech synthesis for role-play scenarios. While our work aims to improve human-computer interaction through more natural and contextually appropriate speech, we acknowledge potential misuse risks. The ability to generate highly expressive and character-consistent speech could be exploited for deceptive purposes, such as impersonation or generating misleading audio content. We advocate for responsible deployment with appropriate safeguards, including watermarking and usage monitoring.

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

# A. Prompts Used in Data Construction

In this section, we provide the specific prompts used in our data construction pipeline, including metadata extraction, scene description generation, and character profile inference.

## A.1. Metadata Extraction Prompt

We employ DeepSeek-R1 to infer the drama series name and episode number from raw YouTube video titles. We utilize a few-shot prompting strategy to ensure the output adheres to a strict JSON format.

**Instruction:** Below is the YouTube video title of a specific episode of a TV series. Please provide the name of the series and the episode number. The output must be in JSON format as follows:

```
{
    "drama_name": "Series Name",
    "episode": "Episode Number"
}
```

**Example 1:**

Input: "<Happiness Sets Sail> Episode 10 Wu Jing resents her father's constraints, Song Honghua agrees to transfer for money, CCTV Drama"

Output:

```
{
    "drama_name": "Happiness Sets Sail",
    "episode": "Episode 10"
}
```

**Example 2:**

Input: "[ENG SUB] Swords of Legends II EP04 (Starring Fu Xinbo, Ying Er, Aarif Rahman, Ken Chang)"

Output:

```
{
    "drama_name": "Swords of Legends II",
    "episode": "Episode 4"
}
```

**Query:** The video title I need you to process is: "{video_title}" Please provide the drama name and episode number. Output strictly in JSON format with no additional text.

**Output:**

## A.2. Scene Annotation Prompt

To generate objective scene descriptions ($\mathcal{S}$), we utilize Qwen-VL-7B. We explicitly instruct the model to ignore dialogue content and focus solely on the physical environment and atmosphere.

**System Prompt:** You are an AI assistant specialized in video understanding, capable of analyzing video segments and generating descriptions.

**User Prompt:** Please describe the scene and environment where the dialogue takes place in this video segment using approximately 25 words. (Ignore the dialogue content; focus solely on the objective description of the physical scene and environment.) `<video>`

### A.3. Character Profiling Prompt

We generate character profiles ($\mathcal{P}$) by feeding the complete dialogue script of an episode into the LLM. The model is tasked with inferring the identity and personality of each speaker based on the interaction history.

> **Instruction:** You are a professional expert in TV drama analysis. I will provide all dialogue information from Episode {episode} of the drama "{drama_name}", including text transcripts and speaker tags.
>
> Please infer the *identity* of each speaker and provide a *character description* for each.
>
> *Note:* The final output must be in a strict, parsable JSON format. The keys for the JSON objects must include: `speaker_id`, `identity`, and `character`.
>
> The dialogue information for "{drama_name}" {episode} is provided below:
>
> ```
> {
>     "segments": [
>         {
>             "text": "How is it? Are you used to living here these days?",
>             "speaker_id": "GLOBAL_SPEAKER_0"
>         },
>         {
>             "text": "It's quite good.",
>             "speaker_id": "GLOBAL_SPEAKER_1"
>         },
>         {
>             "text": "By the way, Mei Jing hasn't been dining with us often...",
>             "speaker_id": "GLOBAL_SPEAKER_1"
>         },
>         ...
>     ]
> }
> ```
>
> Please start your analysis and output the results.

## B. Cross-Lingual Generalization of MCLP

To examine whether MCLP generalizes beyond Mandarin RP-TTS, we conduct a supplementary experiment on a multilingual expressive speech dataset Genshin-Voice[4]. We select 1,000 English and 1,000 Mandarin samples from 20 speakers. For each language, we evaluate MCLP under two zero-shot TTS prompting settings using IndexTTS2 and Spark-TTS: (1) using the target audio itself as the prompt (*self*), and (2) using the preceding audio from the same speaker as the prompt for the current sample (*prev*). Since self-prompted synthesis provides a more style-matched prompt than prev-prompted synthesis, a valid style-consistency metric should assign higher scores to the former.

*Table 4.* Cross-lingual generalization of MCLP. Higher scores indicate stronger stylistic consistency. Self-prompted synthesis consistently receives higher MCLP than prev-prompted synthesis.

| Lang | GT | Resys | IndexTTS2 | | Spark-TTS | |
|------|------|--------|-------|-------|-------|-------|
| | | | self | prev | self | prev |
| en | -0.815 | -2.398 | -3.856 | -4.233 | -4.017 | -4.202 |
| zh | -0.757 | -2.153 | -3.197 | -3.633 | -3.427 | -3.607 |
| all | -0.786 | -2.275 | -3.527 | -3.933 | -3.722 | -3.905 |

*Resys* denotes reconstructing ground-truth audio using the Step-Audio-2 tokenizer. As shown in Table 4, MCLP consistently assigns higher scores to self-prompted samples than to prev-prompted samples across both English and Mandarin, suggesting that MCLP can capture stylistic consistency beyond the Mandarin RP-TTS setting.

---

[4]https://huggingface.co/datasets/simon3000/genshin-voice

## C. Dataset Analysis

We provide additional dataset statistics and fine-grained evaluation results to better characterize the diversity of WenetSpeech-RP-TTS and the robustness of our method across different evaluation scenarios. Specifically, we analyze scene categories, the number of speakers per scene, and model performance across scene types, dialogue lengths, and speaker counts.

### C.1. Scene Category Distribution

Table 5 reports the scene category distribution of the training set and the 900-sample test subset. The dataset covers a broad range of drama genres, including military/war, costume/wuxia, suspense/spy, romance, family/ethics, and urban/modern scenes. This diversity provides varied expressive contexts for evaluating role-play speech style.

*Table 5.* Scene category distribution in the training set and the 900-sample test subset.

| Category | Train (304,489) | | Test Subset (900) | |
|---|---|---|---|---|
| | Count | % | Count | % |
| Military/War | 29,683 | 9.7 | 105 | 11.67 |
| Animation/Anime | 733 | 0.2 | 8 | 0.89 |
| Costume/Wuxia | 51,322 | 16.9 | 173 | 19.22 |
| Comedy | 6,155 | 2.0 | 28 | 3.11 |
| Fantasy | 15,342 | 5.0 | 66 | 7.33 |
| Family/Ethics | 56,527 | 18.6 | 113 | 12.56 |
| Suspense/Spy | 65,165 | 21.4 | 205 | 22.78 |
| Romance | 28,878 | 9.5 | 73 | 8.11 |
| Urban/Modern | 50,268 | 16.5 | 127 | 14.11 |
| Others | 416 | 0.1 | 2 | 0.22 |

### C.2. Speakers Per Scene Distribution

Table 6 shows the distribution of the number of speakers per scene.

*Table 6.* Distribution of the number of speakers per scene.

| #Speakers | Train | | Test Subset | |
|---|---|---|---|---|
| | Count | % | Count | % |
| 1 | 224,348 | 72.8 | - | - |
| 2 | 74,815 | 24.3 | 694 | 77.11 |
| 3 | 8,301 | 2.7 | 170 | 18.89 |
| 4 | 540 | 0.2 | 32 | 3.56 |
| 5 | 19 | 0.0 | 4 | 0.44 |

### C.3. Performance Across Scene Categories

Table 7 reports MCLP scores across scene categories. The proposed model achieves the best performance in all categories, demonstrating that the improvement is not limited to a particular drama genre. This also suggests that MCLP-based alignment is effective across diverse expressive contexts, including suspense, romance, family, fantasy, and military scenes.

### C.4. Performance by Dialogue Length

To evaluate whether the method remains effective under different dialogue-history lengths, we report MCLP scores by the number of dialogue turns in Table 8. The proposed model consistently outperforms all baselines for every dialogue length from 2 to 10 turns.

As shown in Table 8, the proposed model maintains a consistent advantage across all dialogue lengths. The overall trend also suggests that longer dialogue histories can provide richer contextual cues for modeling stylistic continuity.

*Table 7.* MCLP scores by scene category. Best results are in **bold**.

| Category | Count | GPT-Audio | MiMo-Audio | Step-Audio-2 | Proposed |
|---|---|---|---|---|---|
| Military/War | 105 | -4.693 | -4.594 | -4.672 | **-4.477** |
| Animation | 8 | -4.974 | -4.847 | -4.909 | **-4.840** |
| Costume/Wuxia | 173 | -4.756 | -4.642 | -4.735 | **-4.531** |
| Comedy | 28 | -5.064 | -4.971 | -5.049 | **-4.876** |
| Fantasy | 66 | -5.009 | -4.916 | -4.987 | **-4.798** |
| Family/Ethics | 113 | -5.034 | -4.945 | -5.011 | **-4.823** |
| Suspense/Spy | 205 | -4.804 | -4.726 | -4.793 | **-4.594** |
| Romance | 73 | -4.754 | -4.649 | -4.704 | **-4.530** |
| Urban/Modern | 127 | -4.929 | -4.831 | -4.918 | **-4.717** |

*Table 8.* MCLP scores by number of dialogue turns. Best results are in **bold**.

| Turns | Count | GPT-Audio | MiMo-Audio | Step-Audio-2 | Proposed |
|---|---|---|---|---|---|
| 2 | 100 | -4.884 | -4.801 | -4.867 | **-4.721** |
| 3 | 100 | -4.908 | -4.810 | -4.886 | **-4.690** |
| 4 | 100 | -4.975 | -4.882 | -4.945 | **-4.768** |
| 5 | 100 | -4.975 | -4.860 | -4.953 | **-4.750** |
| 6 | 100 | -4.740 | -4.652 | -4.732 | **-4.515** |
| 7 | 100 | -4.961 | -4.853 | -4.935 | **-4.751** |
| 8 | 100 | -4.763 | -4.672 | -4.736 | **-4.528** |
| 9 | 100 | -4.739 | -4.640 | -4.720 | **-4.516** |
| 10 | 100 | -4.695 | -4.607 | -4.686 | **-4.480** |
| All | 900 | -4.849 | -4.753 | -4.829 | **-4.636** |

## C.5. Performance by Number of Speakers

We further evaluate model robustness under different role configurations by grouping test scenes according to the number of speakers. As shown in Table 9, the proposed model achieves the best MCLP scores across all speaker-count groups.

*Table 9.* MCLP scores by number of speakers per scene. Best results are in **bold**.

| #Speakers | Count | GPT-Audio | MiMo-Audio | Step-Audio-2 | Proposed |
|---|---|---|---|---|---|
| 2 | 694 | -4.830 | -4.730 | -4.811 | **-4.614** |
| 3 | 170 | -4.892 | -4.812 | -4.865 | **-4.679** |
| 4 | 32 | -5.011 | -4.911 | -4.990 | **-4.823** |
| 5 | 4 | -5.074 | -5.000 | -5.065 | **-4.937** |

The performance gap remains consistent as the number of speakers increases, indicating that our method can better preserve stylistic consistency even when role interactions become more complex.

