# OpenReview forum: "Evaluating and Rewarding LALMs for Expressive Role-Play TTS via Mean Continuation Log-Probability"
_ICML.cc/2026/Conference — ICML 2026 regular_

### Official Review · Reviewer_UpUi · 2026-03-11

**Soundness:** 2
**Presentation:** 2
**Significance:** 3
**Originality:** 3
**Overall Recommendation:** 5
**Confidence:** 3

**Summary:**

This paper introduces Role-Play TTS, a task where a model must generate speech that is not only faithful to the given text, but also stylistically consistent with scene descriptions, character profiles, and multi-turn dialogue history. To address the lack of a good style-consistency metric for this setting, the paper proposes MCLP, and further uses it as an RL reward together with CER in a GRPO-based training pipeline. The paper also builds a large RP-TTS dataset and reports results against strong audio-language baselines.

**Compliance With Llm Reviewing Policy:**

Affirmed.

**Final Justification:**

This paper introduces Role-Play TTS and proposes MCLP for evaluating style consistency, together with an RL-based training framework and a large RP-TTS dataset. My main concerns were about the reported gains, evaluation, analysis, and dataset details. The authors’ rebuttal addressed these issues well, improved my confidence in the work, and positively changed my evaluation.

**Key Questions For Authors:**

1. Why does the evaluation not include any similarity-related metrics?
2. Have you conducted any human listening tests specifically for the MCLP reward ablation, or do you plan to provide demo-page examples showing different MCLP values and ablation results?
3. Could you provide a finer-grained analysis of both the dataset and the evaluation scenarios? For example, what is the distribution over the number of distinct speakers per scene, and how does performance vary across different dialogue lengths, speaker counts, or scene types?
4. Do you plan to release the dataset, checkpoints, and training/evaluation code? This would make the work much more valuable to the community.

**Limitations:**

No. The paper should more explicitly discuss misuse risks (e.g., deceptive expressive speech generation), data governance issues for drama-derived data, and domain/evaluator limitations of the proposed benchmark and MCLP metric.

**Strengths And Weaknesses:**

Strengths
+ The paper introduces the RP-TTS task in a relatively standardized manner, isolating the research question: "When given text content, how to make it sound like the character within multi-turn contexts." I believe this is a reasonable and important division because it is more focused than free speech agents and closer to the demand for character-based speech generation.
+ Using a pretrained LALM’s continuation likelihood as a proxy for style consistency is novel and conceptually appealing. The paper also provides some evidence that MCLP correlates with human judgments.

Weaknesses
+ My main concern is that the paper does not yet provide sufficiently strong evidence for the practical value of the MCLP reward itself. In the ablation study, the w/o MCLP Reward variant already performs very strongly, and even achieves better CER than the full model, while the gain from adding MCLP mainly appears on the proposed MCLP metric. This makes the incremental benefit of MCLP reward feel somewhat unclear from a perceptual perspective.
+ The experimental analysis is somewhat limited for such a new and complex task. The paper does not include similarity-related metrics, and there is relatively little breakdown analysis across different scenarios, dialogue lengths, or role configurations. Some protocol details are also not fully clear, especially for evaluation

---

> ### Author Rebuttal · Authors · 2026-03-30
>
> We sincerely thank you for your constructive feedback. We would like to address your questions below.
>
> **Q1**: Human listening tests and other evidence for MCLP Reward Ablation.
>
> **A1**: Thank you for your constructive suggestion. To provide a more comprehensive analysis of the MCLP reward, we supplemented subjective MOS and similarity metrics for ablation variants.
>
> Without the MCLP reward, although CER is lower, prosody becomes flatter and expressiveness decreases, weakening stylistic consistency. The human listening results are shown below:
>
> |Model|w. Audio History|w/o. Audio History|All|
> |:-|:-:|:-:|:-:|
> |**Proposed**|**3.682**|**3.342**|**3.576**|
> |Proposed w/o MCLP Reward|2.241|2.528|2.331|
>
> (Full results: **Table in A2 of Response to Reviewer Mdgo**)
>
> The full model achieves higher perceptual style consistency than the variant without the MCLP reward. The similarity-related metrics are reported in **A2**. We also added a column to the anonymous demo page (https://mclp-tts.github.io/mclp/).
>
> **Q2**: Similarity-Related Metrics.
>
> **A2**: Our primary goal is to address the challenge of style consistency. However, existing similarity metrics only partially capture stylistic consistency and do not fully align with human perception. Therefore, we initially focused on human subjective evaluation as the main criterion and did not include similarity-related metrics.
> For a more comprehensive comparison, we have now supplemented the evaluation with speaker similarity (CAM++) and emotional similarity (Emo2Vec) metrics:
>
> |Model||w. Audio History||w/o. Audio History|
> |:-|:-:|:-:|:-:|:-:|
> ||CAM++|Emo2Vec|CAM++|Emo2Vec|
> |GPT-Audio|0.636|0.875|0.635|0.884|
> |MiMoAudio|0.699|0.902|0.698|0.903|
> |StepAudio2|0.629|0.864|0.632|0.864|
> |CosyVoice3|-|-|0.651|0.905|
> |Higgs Audio V2|-|-|0.614|0.856|
> |OV-InstructTTS|-|-|0.669|0.900|
> |Qwen3TTS|-|-|0.630|0.879|
> |**Proposed**|**0.724**|**0.917**|**0.704**|**0.910**|
> |Proposed w/o MCLP Reward|0.687|0.898|0.670|0.886|
> |Proposed w/o WER Reward|0.677|0.914|0.621|**0.910**|
>
> Although these similarity metrics do not perfectly correspond to human subjective judgments, our method achieves the best performance across all metrics.
>
> **Q3**: Dataset and evaluation scenario analysis.
>
> **A3**: Thank you for your suggestion. We provide additional statistics and fine-grained analyses, which will be included in the appendix.
>
> (1) Scene category distribution:
>
> |Category|Train 304,489|Test-sub 900|
> |-|-|-|
> |Military/War|29,683 (9.7%)|105 (11.67%)|
> |Animation/Anime|733 (0.2%)|8 (0.89%)|
> |Costume/Wuxia|51,322 (16.9%)|173 (19.22%)|
> |Comedy|6,155 (2.0%)|28 (3.11%)|
> |Fantasy|15,342 (5.0%)|66 (7.33%)|
> |Family/Ethics|56,527 (18.6%)|113 (12.56%)|
> |Suspense/Spy|65,165 (21.4%)|205 (22.78%)|
> |Romance|28,878 (9.5%)|73 (8.11%)|
> |Urban/Modern|50,268 (16.5%)|127 (14.11%)|
> |Others|416 (0.14%)|2 (0.22%)|
>
> (2) Number of speakers per scene:
>
> |N_spk per Scene|Train|Test-sub|
> |-|-|-|
> |1|224,348 (72.8%)|-|
> |2|74,815 (24.3%)|694 (77.11%)|
> |3|8,301 (2.7%)|170 (18.89)|
> |4|540 (0.2%)|32 (3.56)|
> |5|19 (0.0%)|4 (0.44%)|
>
> (3) Performance across scene categories:
>
> |Category|Count|GPTAudio|MimoAudio|StepAudio2|Proposed|
> |:-:|:-:|:-:|:-:|:-:|:-:|
> |Military/War|105|-4.693|-4.594|-4.672|**-4.477**|
> |Animation|8|-4.974|-4.847|-4.909|**-4.840**|
> |Costume/Wuxia|173|-4.756|-4.642|-4.735|**-4.531**|
> |Comedy|28|-5.064|-4.971|-5.049|**-4.876**|
> |Fantasy|66|-5.009|-4.916|-4.987|**-4.798**|
> |Family|113|-5.034|-4.945|-5.011|**-4.823**|
> |Suspense|205|-4.804|-4.726|-4.793|**-4.594**|
> |Romance|73|-4.754|-4.649|-4.704|**-4.530**|
> |Urban|127|-4.929|-4.831|-4.918|**-4.717**|
> |Others|2|-4.899|-4.838|-4.949|**-4.838**|
>
> Across all scene categories, the proposed model consistently achieves the best performance.
>
> (4) Effect of dialogue length:
>
> |Turns|Count|GPTAudio|MimoAudio|StepAudio2|Proposed|
> |:-:|:-:|:-:|:-:|:-:|:-:|
> |2|100|-4.884|-4.801|-4.867|**-4.721**|
> |3|100|-4.908|-4.810|-4.886|**-4.690**|
> |4|100|-4.975|-4.882|-4.945|**-4.768**|
> |5|100|-4.975|-4.860|-4.953|**-4.750**|
> |6|100|-4.740|-4.652|-4.732|**-4.515**|
> |7|100|-4.961|-4.853|-4.935|**-4.751**|
> |8|100|-4.763|-4.672|-4.736|**-4.528**|
> |9|100|-4.739|-4.640|-4.720|**-4.516**|
> |10|100|-4.695|-4.607|-4.686|**-4.480**|
> |All|900|-4.849|-4.753|-4.829|**-4.636**|
>
> The proposed model consistently outperforms all baselines across different dialogue turns. As the number of turns increases, the MCLP score improves, indicating that more turns improve style consistency.
>
> (5) Effect of the number of speakers per scene.
>
> |N_spk|Count|GPTAudio|MimoAudio|StepAudio2|Proposed|
> |:-:|:-:|:-:|:-:|:-:|:-:|
> |2|694|-4.830|-4.730|-4.811|**-4.614**|
> |3|170|-4.892|-4.812|-4.865|**-4.679**|
> |4|32|-5.011|-4.911|-4.990|**-4.823**|
> |5|4|-5.074|-5.000|-5.065|**-4.937**|
> |All|900|-4.849|-4.753|-4.829|**-4.636**|
>
> The proposed model remains the best-performing method across different numbers of speakers.
>
> **Q4**: Open-source.
>
> **A4**: **All our datasets, checkpoints, and code are ready and will definitely be open-sourced**.

---

> > ### Author Rebuttal · Reviewer_UpUi · 2026-04-02
> >
> > I thank the authors for the detailed response. My concerns have been largely addressed. I hope that the details of the human evaluation, the dataset statistics, and more model results can be included in the final version of the paper. I will raise my score accordingly.

---

> > > ### Author Response · Authors · 2026-04-02
> > >
> > > Thank you very much for your constructive feedback and valuable suggestions. Your suggestions have been very helpful in improving the paper. We will carefully incorporate these improvements into the revised version, including more details on the human evaluation, the dataset statistics, and additional model results. In addition, we will ensure that our datasets, checkpoints, and code are open-sourced to support future research by the community.

---

### Official Review · Reviewer_25yh · 2026-03-12

**Soundness:** 2
**Presentation:** 2
**Significance:** 2
**Originality:** 2
**Overall Recommendation:** 4
**Confidence:** 3

**Summary:**

This paper studies the problem of expressive role-play text-to-speech (RP-TTS) using Large Audio Language Models (LALMs). The authors argue that maintaining stylistic consistency with role-play instructions and dialogue context remains a key challenge for current TTS systems. To address this issue, the paper proposes Mean Continuation Log-Probability (MCLP), an evaluation metric that measures stylistic consistency by computing the average log-likelihood of ground-truth audio tokens conditioned on generated speech using a pretrained LALM. The metric is further used as a reinforcement learning reward signal to improve stylistic alignment in RP-TTS models.

**Compliance With Llm Reviewing Policy:**

Affirmed.

**Final Justification:**

The rebuttal solves some of my concerns, I have update my score to 4.

**Key Questions For Authors:**

1. The proposed MCLP metric relies on a pretrained LALM to compute continuation likelihood. How sensitive are the evaluation results to the choice of the underlying LALM used for computing the metric?

2. Since the proposed metric is also used as a reward for RL optimization, how does the method avoid overfitting to the metric itself? Are there experiments demonstrating generalization beyond the training objective?

**Limitations:**

No. The paper does not provide a clear limitations discussion. Although an impact statement is included, it does not meaningfully analyze the limitations of the proposed metric or the potential risks of reward-based optimization for speech generation.

**Strengths And Weaknesses:**

**Strengths**

1.  The paper studies an interesting problem: improving stylistic consistency in role-play TTS scenarios. Modeling expressive speech in multi-turn conversational contexts is challenging, and the paper highlights the lack of reliable evaluation metrics for style alignment. The idea of using the log-likelihood of ground-truth audio tokens as a proxy for stylistic consistency is conceptually intuitive and provides a potential direction for objective evaluation of expressive speech generation.

2. Another positive aspect is the attempt to integrate the proposed metric into the training pipeline as a reinforcement learning reward. The hybrid reward combining stylistic consistency (MCLP) and content fidelity (CER) is a reasonable design choice for balancing intelligibility and expressiveness. The paper also introduces a relatively large RP-TTS dataset derived from drama videos, which may be useful for future research on expressive TTS.

**Weaknesses**

1. The novelty of the proposed metric appears limited. MCLP essentially computes the log-likelihood of ground-truth tokens conditioned on generated audio using a pretrained LALM. This formulation is closely related to standard likelihood-based evaluation methods and does not clearly demonstrate why it uniquely captures stylistic consistency rather than general acoustic similarity. The theoretical justification based on mutual information is relatively weak and relies on strong assumptions about the pretrained LALM acting as an accurate density estimator.

2. The evaluation methodology raises concerns about potential circularity. The proposed metric relies on the probability distribution of a pretrained LALM, which is also closely related to the modeling framework used for generation. As a result, improvements in MCLP may reflect biases of the underlying model rather than true improvements in stylistic quality. The paper does not analyze this potential bias or compare the metric against alternative style evaluation methods.

3. The experimental validation is limited to the RP-TTS setting. While the results show improvements on this task, it remains unclear whether the proposed metric generalizes to broader speech generation tasks such as general expressive TTS, voice acting synthesis, or conversational speech generation.

4. The paper introduces a new dataset but provides limited discussion of dataset quality and annotation reliability. For example, scene and character descriptions are generated using large language models, which may introduce noise or bias in the annotations. The impact of such noise on training and evaluation is not discussed.

5. The paper does not provide a clear limitations discussion. Although an impact statement is included, it does not meaningfully analyze the limitations of the proposed metric or the potential risks of reward-based optimization for speech generation.

---

> ### Author Rebuttal · Authors · 2026-03-30
>
> We sincerely thank you for your constructive feedback. We would like to address your questions and clarify your concerns.
>
> **Q1**: Why does MCLP uniquely capture stylistic consistency rather than general acoustic similarity? How sensitive is it to the choice of the underlying LALM?
>
> **A1**:  We thank you for the insightful question. Although MCLP is likelihood-based, it incorporates several carefully designed constraints to bias the metric toward stylistic consistency rather than general acoustic similarity.
>
> (1) **Motivation: leveraging generative audio language modeling.**
> Generative LALMs naturally exhibit zero-shot style cloning ability. This property has been widely exploited in LALM-based TTS systems. Therefore, speech with a consistent style provides a more suitable context for continuation prediction, leading to higher MCLP scores. This also explains why models without generative audio pretraining (e.g., Qwen-Omni) are less suitable for computing MCLP.
>
> (2) **Tokenizer choice to suppress acoustic information.**
> We use a single-codebook tokenizer from StepAudio2 that encodes semantic and stylistic information while minimizing acoustic factors. In contrast, LALMs based on multi-layer RVQ tokenizers (e.g., MiMoAudio) mix in more substantial acoustic information.
>
> (3) **Carefully designed context for MCLP inference.**
> To suppress textual influence, MCLP uses dual-turn inputs with **identical transcripts** and computes likelihood only on continuation speech tokens in the interleaved sequence, where text tokens are teacher-forced.
>
> Therefore, the MCLP formulation and model choice are all carefully designed.
>
> **Q2**: Compare the metric against alternative style evaluation methods.
>
> **A2**: Robust objective metrics for style consistency remain limited. Therefore, human subjective evaluation is the primary assessment in our study. We have verified the alignment between MCLP and MOS scores (Fig. 5), showing that MCLP correlates well with human judgments.
>
> Following reviewer Mdgo and UpUi's suggestions, we further expanded the subjective evaluation to provide a more comprehensive comparison (**Table in A2 of Response to Reviewer Mdgo**). These results confirm consistent improvements in perceptual style consistency.
>
> To compare with conventional approaches, we also report similarity-based metrics CAM++ for speaker and Emo2Vec for emotion (**Table in A2 of Response to Reviewer UpUi**). These similarity metrics do not always align with MOS, which highlights the limitation of traditional similarity metrics for evaluating stylistic consistency and motivates the need for MCLP. Our method still achieves the best performance across all metrics, demonstrating generalization beyond the training objective.
>
> **Q3**: Generalization of MCLP.
>
> **A3**: To evaluate the generalization of MCLP, we conducted a small experiment measuring general speech style consistency. Using zero-shot TTS, we constructed pairs of samples with relatively similar and dissimilar styles under two prompting settings. We evaluated these pairs on a multilingual expressive dataset (Chinese and English). Across both languages, samples with stronger stylistic similarity consistently receive higher MCLP scores, indicating that MCLP captures general style consistency beyond the RP-TTS setting (see **Table in A3 of Response to Reviewer Mdgo**).
> In addition, our RP-TTS dataset itself covers diverse scene categories. We provide detailed statistics and analyses of scene types in **A3 of Response to Reviewer UpUi**, demonstrating broad coverage across expressive speech scenarios.
> These results suggest that MCLP has good generalization as a style consistency metric. We believe it can be applied to broader tasks such as general expressive TTS. Future work will further explore using MCLP as a reward for training models on these tasks.
>
> **Q4**: Dataset quality and annotation noise.
>
> **A4**: As shown in Table 1, we filtered 311,938 scenes from 598,701 candidates, removing inconsistent or low-quality samples.
> To further assess annotation quality, we conducted a manual inspection of 1,000 randomly sampled scenes. We checked whether scene descriptions and character roles were consistent with the corresponding speech. Among them, 932 samples passed the inspection, indicating that the annotations largely meet the intended quality standard.
> Some minor inconsistencies may still exist in attributes such as speaker or character naming, but these have a limited impact on our task. As long as the scene description and speech remain aligned within each ~30-second segment, the data still provides useful supervision for style consistency modeling.
>
> We will open-source the dataset to enable continuous use and further improvement.
>
> **Q5**: Limitations.
>
> **A5**: We will supplement the limitations and impact statements in the revision:
> - Single-language dataset
> - Further validation across more tasks
> - Noise from automatic annotation
> - Potential misuse risks

---

> > ### Author Rebuttal · Reviewer_25yh · 2026-04-01
> >
> > I thank the authors for the detailed rebuttal and the supplementary experiments.  My concerns have been adequately addressed. I will update my score.

---

> > > ### Author Response · Authors · 2026-04-02
> > >
> > > Thank you very much for your constructive feedback and valuable suggestions. Your suggestions have helped us substantially strengthen the paper. We will carefully incorporate these improvements into the revised version, including a clearer discussion of the choice of the underlying LALM, comparisons with additional metrics, further discussion of generalization and dataset quality, and a more explicit limitations section.

---

### Official Review · Reviewer_Mdgo · 2026-03-13

**Soundness:** 2
**Presentation:** 3
**Significance:** 3
**Originality:** 3
**Overall Recommendation:** 5
**Confidence:** 4

**Summary:**

1. This paper tackles the problem of maintaining stylistic consistency in Role-Play Text-to-Speech, where synthesized speech must match scene descriptions and character profiles across multi-turn dialogues.
2. The authors propose Mean Continuation Log-Probability (MCLP). MCLP measures how likely a ground-truth audio continuation is given a candidate generated audio, serving as a proxy for style similarity in the LALM's latent space.
4. Beyond evaluation, MCLP is used as a reward signal within a Group Relative Policy Optimization (GRPO) framework, combined with a Character Error Rate (CER) penalty to prevent reward hacking.
5. The authors also construct a large-scale RP-TTS dataset (~311k scenes, 1,435 hours) derived from Chinese drama videos with scene and character annotations.

**Compliance With Llm Reviewing Policy:**

Affirmed.

**Final Justification:**

Rebuttal addressed my concerns

**Key Questions For Authors:**

Check weaknesses

**Limitations:**

Key limitations that should be acknowledged include: the single-language experimental scope, reliance on proprietary model components and the small scale of subjective evaluation.

**Strengths And Weaknesses:**

Strengths:

1. Well-motivated and principled metric. The core idea of using continuation log-probability from a pretrained LALM as a style consistency measure is elegant and well-grounded. The information-theoretic justification connecting MCLP to mutual information between generated and ground-truth audio (with content held constant) is sound.

2.  Strong experimental validation. The proposed method achieves large improvements over competitive baselines across all metrics. The subjective MOS of 3.646 substantially exceeds the best baseline (MiMo-Audio at 2.471) and approaches ground truth (4.411).

3. Comprehensive dataset construction. The resulting dataset is substantial and fills a genuine gap for RP-TTS research.

Weaknesses:

1. Narrow baseline comparison. The baselines are limited to three systems, and the comparison is somewhat uneven, GPT-Audio and MiMo-Audio are general-purpose models not specifically designed for RP-TTS, while the proposed method fine-tunes and RL-aligns on the task-specific dataset. A fairer comparison would include other controllable TTS systems discussed in the related work. This is the biggest limitation.
2. Subjective evaluation scale is limited. Only 31 samples were evaluated by human annotators, and only 30 responses were retained after quality filtering. For a task centered on perceptual quality, this is a small evaluation set. Additionally, the annotators were instructed to ignore timbre and audio quality, but it is unclear how well they could isolate "stylistic consistency" from these confounding factors in practice.
3. Limited language and domain scope. All experiments are conducted exclusively on Chinese drama data. It is unclear how well MCLP generalizes to other languages, domains (e.g., audiobooks, games, customer service), or speaking styles beyond dramatic performance. The authors do not discuss this limitation.

---

> ### Author Rebuttal · Authors · 2026-03-30
>
> We sincerely thank you for your constructive feedback. We would like to address your questions below.
>
> **Q1**: A fairer comparison would include other controllable TTS systems discussed in the related work.
>
> **A1**: We sincerely thank you for your constructive suggestion. We initially selected recent LALM-based systems capable of multi-turn understanding and generation as baselines, as RP-TTS involves multi-turn audio history and scene-aware role conditioning.
>
> However, under a single-turn setting without audio history, it is feasible to include controllable TTS systems for comparison. Therefore, we additionally evaluated four recent controllable TTS systems by rewriting prompts to match each model’s input format for fair comparison. The results are shown below:
>
> |Model|CER (%)|Pinyin WER (%)|MCLP|MOS|
> |:-|:-:|:-:|:-:|:-:|
> |CosyVoice3|4.638|2.938|-4.782|2.350|
> |Higgs Audio V2|3.250|1.728|-4.827|1.750|
> |OV-InstructTTS|7.188|5.597|-4.768|2.864|
> |Qwen3TTS|5.585|3.672|-4.799|2.036|
> |**Proposed**|**1.625**|**0.970**|**-4.687**|**3.342**|
>
> Our method consistently outperforms these controllable TTS baselines across both objective and subjective metrics. We will update the experimental section in the revised paper to include these additional comparisons.
>
> **Q2**: It is unclear how well annotators could isolate “stylistic consistency”; the subjective evaluation scale is limited.
>
> **A2:**  (1) To ensure annotators correctly distinguished stylistic consistency from other factors, we inserted quality-control samples during evaluation. These samples were used to identify inattentive or inconsistent annotations and were excluded from the final statistics. After filtering, two annotators were removed from the total pool.
>
> (2) The 95% CIs reported in Table 2 are very small (±0.052 / ±0.063 / ±0.074 / ±0.058 / ±0.060). We further performed statistical significance tests on the subjective scores. We compared the proposed method against each baseline using one-sided Wilcoxon signed-rank tests on raw ratings with Holm correction, and report rank-biserial correlation (RBC) as the effect size.
>
>   - vs GPT: p_holm < 1e-6, RBC = 0.962
>   - vs MiMoAudio: p_holm < 1e-6, RBC = 0.804
>   - vs StepAudio2: p_holm < 1e-6, RBC = 0.977
>
> The results confirm that the improvements of our method are statistically significant.
>
> (3) Besides, following your suggestion and Reviewer UpUi’s feedback, we additionally conducted a more comprehensive subjective evaluation. This study includes:
>
> - the four controllable TTS baselines added in **Q1**
> - ablation models
> - all original models
>
> To ensure fairness, all models were re-evaluated jointly under the same setup. The results are shown below:
>
> | Model | w. Audio History | w/o. Audio History | All  |
> |:---|:---:|:---:|:---:|
> | GT                     | 4.524 | 4.322 | 4.461 |
> | GPT-Audio        | 1.869 | 2.017 | 1.915 |
> | MiMoAudio       | 2.410 | 2.647 | 2.484 |
> | StepAudio2       | 1.924 | 1.706 | 1.856 |
> | CosyVoice3       | -        | 2.350 | 2.350 |
> | Higgs Audio V2 | -        | 1.750 | 1.750 |
> | OV-InstructTTS | -        | 2.864 | 2.864 |
> | Qwen3TTS        | -        | 2.036 | 2.036 |
> | **Proposed** | **3.682** | **3.342** | **3.576** |
> | Proposed w/o MCLP Reward | 2.241 | 2.528 | 2.331 |
> | Proposed w/o WER Reward   | 1.187 | 1.053 | 1.145 |
>
> **Q3**: Limitations (language and domain scope).
>
> **A3:**  The current evaluation is limited to Chinese primarily due to the lack of available datasets for the RP-TTS task in other languages.
>
> We supplemented a simple experiment to evaluate the generalization of MCLP on English. We used a multilingual expressive dataset (genshin-voice), selecting 1000 English and 1000 Chinese samples from 20 speakers. Under two settings, we synthesized speech using two zero-shot TTS systems:
> (1) using each audio itself as the prompt;
> (2) using the preceding audio from the same speaker as the prompt for the subsequent sample.
>
> |Lang|GT|Resys|IndexTTS2 self|IndexTTS2 prev|Spark-TTS self|Spark-TTS prev|
> |:-:|:-:|:-:|:-:|:-:|:-:|:-:|
> |en|-0.815|-2.398|-3.856|-4.233|-4.017|-4.202|
> |zh|-0.757|-2.153|-3.197|-3.633|-3.427|-3.607|
> |all|-0.786|-2.275|-3.527|-3.933|-3.722|-3.905|
>
> Resys denotes reconstructing ground-truth audio using the StepAudio2 tokenizer. Self-prompted synthesis naturally exhibits stronger stylistic consistency. Across all languages and settings, MCLP consistently assigns higher scores to samples with greater stylistic consistency, demonstrating promising cross-lingual generalization.
>
> Regarding domain scope, the proposed RP-TTS dataset covers diverse scene categories (e.g., military, romance, suspense, comedy, fantasy, urban, etc.), providing substantial domain diversity. Detailed statistics and analyses are provided in our response A3 to Reviewer UpUi.
>
> To facilitate further research, we will open-source the dataset, code, and model checkpoints, enabling future multilingual and cross-domain extensions.

---

> > ### Author Rebuttal · Reviewer_Mdgo · 2026-04-04
> >
> > I thank the authors for the detailed rebuttal and the supplementary experiments. My concerns have been adequately addressed. I will update my score.

---

> > > ### Author Response · Authors · 2026-04-04
> > >
> > > Thank you very much for your constructive feedback and valuable suggestions. Your suggestions have been very helpful in improving the paper. We will carefully incorporate these improvements into the revised version, including comparisons with additional controllable TTS baselines, more details on the human evaluation, and a more explicit discussion of the limitations.

---

### Decision · Program_Chairs · 2026-04-30

**Decision:**

Accept (regular)

**Comment:**

This paper tackles the challenge of maintaining stylistic consistency in Role-Play Text-to-Speech (RP-TTS) models across multi-turn dialogues. The authors introduce Mean Continuation Log-Probability (MCLP), a novel metric leveraging pre-trained Large Audio Language Models (LALMs) to quantify stylistic consistency. Furthermore, they incorporate MCLP as a reinforcement learning reward signal to enhance style alignment and introduce a large-scale RP-TTS dataset (~311k scenes, 1,435 hours) derived from Chinese drama videos.

The reviewers were broadly appreciative of the paper's core motivations. They highlighted the novelty and elegance of the MCLP metric, the strong experimental results, and the high value of the introduced dataset for the RP-TTS community. Following the rebuttal, all three reviewers explicitly acknowledged that their concerns were completely resolved, leading to raised/solidified scores. The paper presents a technically solid methodology, an elegant metric, and a highly valuable dataset. Given the unified positive consensus and the comprehensive author response, I recommend accepting this paper.